# Human papillomavirus genotype distribution among women with and without cervical cancer: Implication for vaccination and screening in Ghana

Yvonne Nartey[1]*, Kwabena Amo-Antwi[2], Philip C. Hill[3], Edward T. Dassah[2], Richard H. Asmah[4], Kofi M. Nyarko[5], Ramatu Agambire[6], Thomas O. Konney[2], Joel Yarney[7], Nelson Damale[8], Brian Cox[9]

1 Department of Adult Health, School of Nursing and Midwifery, University of Ghana, Accra, Ghana, 2 School of Medicine and Dentistry, Kwame Nkrumah University of Science & Technology/Komfo Anokye Teaching Hospital, Kumasi, Ghana, 3 Centre for International Health, Department of Preventive and Social Medicine, Dunedin School of Medicine, University of Otago, Dunedin, New Zealand, 4 Department of Biomedical Sciences, University of Health & Allied Sciences, School of Basic and Biomedical Sciences, Ho, Volta region, Ghana, 5 Disease Control and Prevention Department, Ghana Health Service, Accra, Ghana, 6 Department of Nursing, Garden City University College, Kumasi-Ghana, Ghana, 7 National Centre for Radiotherapy and Nuclear Medicine, Korle Bu Teaching Hospital, Accra, Ghana, 8 Department of Obstetrics and Gynaecology, Korle Bu Teaching Hospital, Accra, Ghana, 9 Hugh Adam Cancer Epidemiology Unit, Department of Preventive and Social Medicine, Dunedin School of Medicine, University of Otago, Dunedin, New Zealand

☯ These authors contributed equally to this work.
* yvonnenartey69@gmail.com, ynartey@ug.edu.gh

## Abstract

### Introduction

Determining the high-risk human papillomavirus (HR-HPV) genotypes burden in women with and without cervical cancer afford a direct comparison of their relative distributions. This quest is fundamental to implementing a future population-based cervical cancer prevention strategy in Ghana. We estimated the cervical cancer risk by HPV genotypes, and the HPV vaccine-preventable proportion of cervical cancer diagnosed in Ghana.

### Materials and methods

An unmatched case-control study was conducted at the two largest cervical cancer treatment centres in Ghana from 1st October 2014 to 31st May 2015. Cases were women diagnosed with cervical cancer and controls were women without cervical cancer who were seeking care at the two hospitals. Nested multiplex polymerase chain reaction (NM-PCR) was used to detect HPV infection in cervical samples. Logistic regression was used to determine the association between the risk of cervical cancer and identified HPV infection. P ≤0.05 was considered statistically significant.

### Results

HPV deoxyribonucleic acid (DNA) data were analysed for 177 women with cervical cancer (cases) and 201 without cancer (controls). Cervical cancer was diagnosed at older ages

**Data Availability Statement:** Data cannot be shared publicly because of ethical concerns. Data will be available from the Ghana Health Service

Ethics Committee (contact via ethics. research@ghsmail.org) for researchers who meet the criteria for access to confidential data.

**Funding:** The authors received no specific funding for this work. However, The Department of Preventive and Social Medicine and the Directors' Cancer Research Trust provided funding for HPV DNA testing. The funders had no role in study design, data collection and analysis, decision to publish, or preparation of the manuscript.

**Competing interests:** The authors have declared that no competing interests exist.

compared to the age at which controls were recruited (median ages, 57 years vs 34 years; p < 0.001). Most women with cervical cancer were more likely to be single with no formal education, unemployed and less likely to live in metropolitan areas compared to women without cervical cancer (all p-value <0.001). HPV DNA was detected in more women with cervical cancer compared to those without cervical cancer (84.8% vs 45.8%). HR-HPV genotypes 16, 18, 45, 35 and 52 were the most common among women with cervical cancer, while 66, 52, 35, 43 and 31 were frequently detected in those without cancer. HPV 66 and 35 were the most dominant non-vaccine genotypes; HPV 66 was more prevalent among women with cervical cancer and HPV 35 in those without cervical cancer. Cervical cancer risk was associated with a positive HPV test (Adjusted OR (AOR): 5.78; 95% CI: 2.92–11.42), infection with any of the HR-HPV genotypes (AOR: 5.56; 95% CI: 3.27–13.16) or multiple HPV infections (AOR: 9.57 95% CI 4.06–22.56).

## Conclusion

Women with cervical cancer in Ghana have HPV infection with multiple genotypes, including some non-vaccine genotypes, with an estimated cervical cancer risk of about six- to ten-fold in the presence of a positive HPV test. HPV DNA tests and multivalent vaccine targeted at HPV 16, 18, 45 and 35 genotypes will be essential in Ghana's cervical cancer control programme. Large population-based studies are required in countries where cervical cancer is most prevalent to determine non-vaccine HPV genotypes which should be considered for the next-generation HPV vaccines.

## Introduction

Cervical cancer is the second common female cancer in Ghana [1]. However, like most low- and middle-income countries (LMICs) access to and uptake of cervical cancer screening and human papillomavirus vaccination are limited in Ghana. Additionally, delayed diagnosis leads to poorer oncologic outcomes, which disproportionately impedes cervical cancer control in Ghana [2, 3]. The recognition that persistent HR-HPV infection causes cervical cancer [4] has engendered a global strategy for cervical cancer prevention in the twenty-first century [5]. The detection of the HPV DNA has become an attractive approach to identifying women at risk of developing cervical cancer. The aetiologic role of HPV of the alpha genus is well documented in several epidemiological studies [6]. The alpha HPVs are classified based on phylogenetic similarity, and their association with cervical intraepithelial lesions and carcinoma [7]. The genotypes 16, 18, 31, 33, 35, 39, 45, 51, 52, 56, 58, 59, 66 and 68, are designated high-risk or carcinogenic to humans. Other genotypes, including 26, 53, 69, 73 and 82, have been suggested as probable high-risk group. The low-risk HPV, HPV-42, -43, -6/11 and -44 [8], cause benign hyperproliferative epithelial lesions [9]. In regions of established cervical cancer programs, HPV vaccination given prior to wide HPV exposure prevents about 90% of targeted HPV-related infections, cervical precancer lesions and cancer [10, 11].

In the last 20 years, major efforts have been made worldwide to generate epidemiological data on cervical HPV-DNA. HPV prevalence and genotype distribution are discrepant, mainly caused by several factors, including early sexual debut, risky sexual habits and lack of systematic vaccination and screening programmes [12]. In West Africa (including Ghana), while infections with genotypes 16, 58, 18, 35 and 52 are common in women with normal cytology,

types 16, 18, 45, 59 and 35 are most prevalent in women with cervical cancer [13]. Apart from the infection with types 16 and 18, significant variation exists in the distribution of the other HR-HPV genotypes (31, 33, 35, 45, 52, and 58) which contributes up to 30% of the cause of invasive cervical cancer worldwide. Of the risk factors, multiple HPV infections and co-infection with HIV are associated with the highest risk of invasive cervical cancer [14]. The prevalence of HPV DNA positivity has individual and community level significance regarding cervical cancer control. The HPV infection profile in different geographical areas has implications for region-specific HPV DNA testing, vaccine choice, and expected efficacies.

The available HPV vaccines have activity against up to nine HPV types (6, 11, 16,18, 31 33 45 52 58) [15]. The bivalent vaccines and later the quadrivalent vaccine, were introduced in Ghana in the last decade, and from 2013 to 2015, Global Alliance for Vaccines and Immunization (GAVI) supported an HPV demonstration project among 10–14-year-old girls in four regions [16]. The geographic HPV genotype variability has increased the debate about HPV vaccine efficacy in the sub-Saharan [17]. The licensed vaccines were developed based on the virus-like particle of the major papillomavirus protein L1, which is highly genotype-specific [18]. Expectedly, limitations exist in vaccine efficacies against non-vaccine alpha HPV type and the level of cross-protection afforded by these vaccines. More recently, cytologic tests have been replaced by polymerase chain reaction (PCR)-based assays for the presence of HR-HPV in a number of high-income countries [19].

Cervical cancer prevention through vaccination against HPV and early detection and treatment of cervical cancer is critical in the WHO strategy for the disease elimination [20]. The female population in Ghana may be exposed to different levels of risk and time to cervical cancer progressions. Determining HR-HPV genotype distribution in women with or without cervical cancer afford a direct comparison of their relative burden and an estimate of the likely progression into invasive cervical cancer. This quest is fundamental to implementing a population-based cervical cancer preventive strategy in Ghana. We estimated the cervical cancer risk by HPV genotype and the HPV vaccine-preventable proportion of cervical cancer in women seen at Ghana's two largest cancer treatment centres.

## Material and methods

### Study design

A case-control study was conducted from October 2014 to May 2015 among women seeking care at Ghana's two largest tertiary hospitals.

### Setting

In the absence of a population-based cervical cancer preventive strategy, several health facilities, including Korle Bu Teaching Hospital (KBTH), Accra and Komfo Anokye Teaching Hospital (KATH), Kumasi, provide opportunist screening for cervical cancer in Ghana. Albeit cervical cancer screening in Ghana is limited, government and non-governmental institutions have actively undertaken projects on cervical cancer. Opportunistic screening for cervical cancer at the two tertiary hospitals started in May 2004 through a John Hopkins Programme for International Education in Obstetrics and Gynecology (JHPIEGO) initiative supported by the Ghana Government and Ministry of Health (MOH). They account for the most significant number of women screened per institution and remain relevant in decisions regarding cervical cancer control in Ghana. The Departments of Obstetrics and Gynaecology and Radiation Oncology of these hospitals provide services for up to 600 women with cervical cancers annually [2].

## Recruitment of study participants

Women with or without cervical cancer aged at least 18 years who resided in Ghana for at least three years prior to the study were eligible for inclusion. Cases were women diagnosed with cervical cancer seeking treatment at the two hospitals within the study period. Within the same period, women who did not have cervical cancer but visited the Obstetrics and Gynaecology departments of the two hospitals for other reasons were recruited as controls. Pregnant women or those previously treated for cervical cancer were excluded. Women on hormonal contraceptive methods were excluded from the controls. Cases were recruited consecutively until the desired sample size was attained. For every case recruited, the next eligible control was contacted for inclusion into the study. The cases and controls were recruited by two nurses working at the Radiation and Gynaecologic Oncology clinics at the two tertiary hospitals.

## Interviews and DNA sample collection

Two specialist gynaecologists and oncology nurses working at gynaecologic and radiation oncology clinics were trained to obtain informed consent, administer a structured questionnaire, and collect cervical samples. After providing informed consent, a questionnaire covering demographic characteristics, sexual behaviour, reproductive and contraceptive history, genital hygiene, and screening history was administered in English or vernacular (mainly Ga or Twi). The interview and cervical sampling for newly diagnosed cases or the controls were done on the same day, while arrangements were initiated to retrieve paraffin-embedded slides for cases with a histology-confirmed diagnosis.

Cervical sampling was done in the minor procedure theatre (with a capacity for examination under anaesthesia). Participants were examined in the lithotomy position on a gynaecologic table. Under a good light source, a sterile bivalve speculum without a lubricant was inserted into the vagina to visualize the cervix or the cervical tumour. For women with obvious cervical tumour, a cytobrush was gently dabbed at several points on the cervical tumour to ensure adequate exfoliated cervical cancer cells were sampled. The cells were recovered into a pre-labelled tube containing DNAgard solution (Biometrica, San Diego, USA) for DNA preservation at room temperature until DNA extraction.

A cervical punch biopsy was taken in suspected cervical cancer cases with reference to local protocols. As a precaution, a vaginal pack impregnated with silver nitrate or acetoacetic acid was prepared and ready to be applied in the event of provoked vaginal bleeding after a punch biopsy. Paraffin-embedded slides were requested from the reporting laboratories for women with a histology-confirmed cancer diagnosis. In the control subjects, exfoliated cells from the ectocervix and endocervix were obtained using spatula and cytobrush, respectively (Pap Pak® cytology kit, Medical Packaging Corporation, Camarillo, CA, USA), with adherence to manufacturer's instructions. The samples were smeared uniformly on a pre-labelled slide, and immediately fixed with a mixture of 95% ethanol and 5% polyethylene glycol (carbowax) (BD-TriPath Imaging). The cells remaining on the spatula and the cytobrush were again by washing them into DNAgard solution (Biometrica, San Diego, USA). The biopsy specimens and slides, for the histologic and cytologic assessments, were sent to the pathology department of the respective institution. The Pap results for the control subjects were reported according to Bethesda nomenclature [21], and any precancer lesion was managed according to the World Health Organization (WHO) protocol [22]. The cervical samples in the DNAgard solution (Biometrica, San Diego, USA) and paraffin-embedded samples were analysed for the presence of HPV DNA at the Department of Medical Laboratory Sciences, Molecular Biology laboratory, School of Biomedical and Allied Health Sciences, University of Ghana, Korle-Bu, Accra.

## HPV DNA purification, amplification, and detection

HPV DNA isolation was performed with a commercially available kit (Qiagen Ltd, Maryland, USA), following manufacturer's instructions for the samples stored in the DNAgard solution (Biometrica, San Diego, USA).

For the paraffin-embedded samples, DNA was extracted as using a method described by Lagheden et al. [23] as follows; briefly; one hundred and eighty (180 μL) ATL buffer from a Qiagen DNA extraction kit (Qiagen Ltd, Maryland, USA) was added to the tube and high-heat treated in 120˚C for 20 min to melt the paraffin. Within the 5 first minutes the tubes were mixed by tapping the tube to make sure that all of the paraffin was under the surface. After 20 min the samples were incubated at room temperature for 3 minutes, followed by a quick centrifugation. 20μL proteinase K was then added, briefly vortex and incubated in 65˚C for 16 hr. The tubes were quickly centrifuged. A solution of 200μL buffer AL (Qiagen Ltd, Maryland, USA) was added and 400μL ethanol, per sample, was prepared. The mixture was added to a DNeasy Mini spin column (Qiagen Ltd, Maryland, USA) and centrifuged 1 min at 8000 rpm. The following steps are performed according to the manufacturer's protocol except for the volume in the elution step that was changed to 25μL AE buffer (Qiagen Ltd, Maryland, USA).

HPV-DNA detection and identification of the genotypes were carried out by nested multiplex PCR (NMPCR) [23]. Primers for the identification of high-risk genotypes 16, 18, 31, 33, 35, 39, 45, 51, 52, 56, 58, 59, 66, and 68 and low-risk genotypes 6/11, 42, 43, and 44 were used. The primers were used in four cocktails, each containing four to five different primer pairs. A single consensus forward primer (GP-E6-3F) and two consensus back primers (GP-E7-5B and GP-E7-6B) were used. The PCR reaction mix of 25μl contained 10X PCR buffer (Promega, USA), 2.5 mM $MgCl_2$, 200μM of each of the four oligonucleotide triphosphates (dNTP) (Promega, USA), 15pmols of each E6/E7 consensus primers and 1.25 units of Taq polymerase enzyme (Promega, USA). Four microlitres (4μl) of DNA extracts was used as template for the amplification reactions for samples extracted from cervical cells and 2μl for DNA from paraffin embedded samples. The PCR was carried out using a PE Applied Biosystems 9700 thermal cycler.

The cycling parameters for the first round PCR with E63F/E75B/E76B consensus primers were as follows: 94˚C for four minutes (initial denaturation), followed by 40 cycles of 94˚C for one minute (denaturation), 40˚C for two minutes (annealing), 72˚C for two minutes (extension) and a single final elongation step of 72˚C for 10 minutes. In the second round PCR, 1μl of first round PCR product, 15pmols of forward and reverse primers for genotyping were used. The other parameters that were used in the first round PCR mix were the same. The cycling parameters were as follows: 94˚C for four minutes followed by 35 cycles of 94˚C for 30 seconds, 56˚C for 30 seconds, 72˚C for 45 seconds and a single final elongation step of 72˚C for four minutes [24]. Plasmid DNA for HPV 16 and 18 were used as a positive control and the negative control was nuclease free water in all experiments for quality control.

The amplification products were analysed by gel electrophoresis on 2% agarose gel and stained with 0.5μg/ml ethidium bromide. Ten microlitres of each sample was added to 2μl of orange G (5X) gel loading dye for the electrophoresis. A one hundred base pair DNA molecular weight marker (Sigma, MO, USA) was run alongside the PCR products. The gel was prepared and electrophoresed in 1X TAE buffer using a mini gel system at 100 volts for one hour and the gel photographed over a UV transilluminator [24].

## Sample size

Overall HPV DNA positivity and multiple HPV infections are critical determinants of cervical cancer risk. HPV prevalence is highest among HIV seropositive women, 75% [25], and lowest,

32.3% [12], in women with rural residence. At a 95% confidence and 80% power, if the true odds ratios in the highest- and lowest-at-risk populations for cervical cancer were assumed to be 2.83 and 1.6 [26], respectively, a sample size of 140 is required to study the risk of cervical cancer in the exposed relative to the unexposed populations. The ratio of controls to cases was 1:1. Allowing 10% for contingency, inappropriate and nonresponses, our estimated sample size was 154 in each group.

## Data analysis

Participants' demographic and HPV infection profiles were summarised using tables, proportions, means and standard deviations, and medians with interquartile ranges (IQR). For the purpose of the analysis, fourteen HPV types (16, 18, 31, 33, 35, 39, 45, 51, 52, 56, 58, 59, 66 and 68) were considered high-risk oncogenic types [27]. Four HPV types (6/11, 42, 43 and 44) were considered low-risk oncogenic types (HPV types 6 and 11 were combined). Participants were classified as HPV DNA positive or negative according to whether HPV DNA was detected in cervical samples. The HPV infection detected was further characterised as a single or multiple HPV infections. HPV DNA prevalence was calculated as a percentage of women positive for any HPV type relative to the total number of women with a negative or positive HPV test. HPV DNA positivity was calculated separately for cases and controls, with their respective denominators. Detected HPV types were dichotomised into vaccine types and non-vaccine types to assess the potential efficacy of the approved HPV vaccines. Odd ratios (ORs) and 95% confidence interval (CI) for the association between risk of cervical cancer and HPV DNA positivity, risk group and infection type were estimated using logistic regression. All analysis was conducted in STATA® version 16.1 (STATA Corporation, College Station, TX). $P \leq 0.05$ was considered statistically significant.

## Ethics

The study was approved by the Committee on Human Research, Publications and Ethics (CHRPE), Kwame Nkrumah University of Science and Technology and KATH (CHRPE/AP/661/19), the Ethics Committees of the University of Otago, New Zealand (Ref: H13/113) and Ghana Health Service (Protocol ID: GHS-ERC: 01/05/14). Written informed consent were obtained from all participants. Respondents were informed in a Ghanaian language (often Twi and Ga) or English about their rights of voluntary participation and withdrawal from the study. They were aware that deciding not to participate in the study would not affect their care at these facilities. The anonymity of the data was assured.

## Results

### Sociodemographic characteristics

Two hundred and thirty (261) cervical cancer cases were seen over the study period at the two centres, of whom 84 were excluded for various reasons, leaving 177 for analysis (Fig 1). Of the 267 potential controls contacted for inclusion into the study, 201 were included in the final analysis, while 66 were excluded (Fig 1). Hence, HPV DNA data were analysed for 177 women with cervical cancer (cases) and 201 without cancer (control). The sociodemographic characteristics of the cases and control are compared in Table 1. Women with cervical cancer were diagnosed at older ages compared to the age at which controls were recruited into the study (median ages, 57 years vs 34 years; p < 0.001). Women with cervical cancer were more likely to be separated/divorced/widowed or unemployed and less likely to live in the metropolitan area or attain secondary or tertiary education compared to women

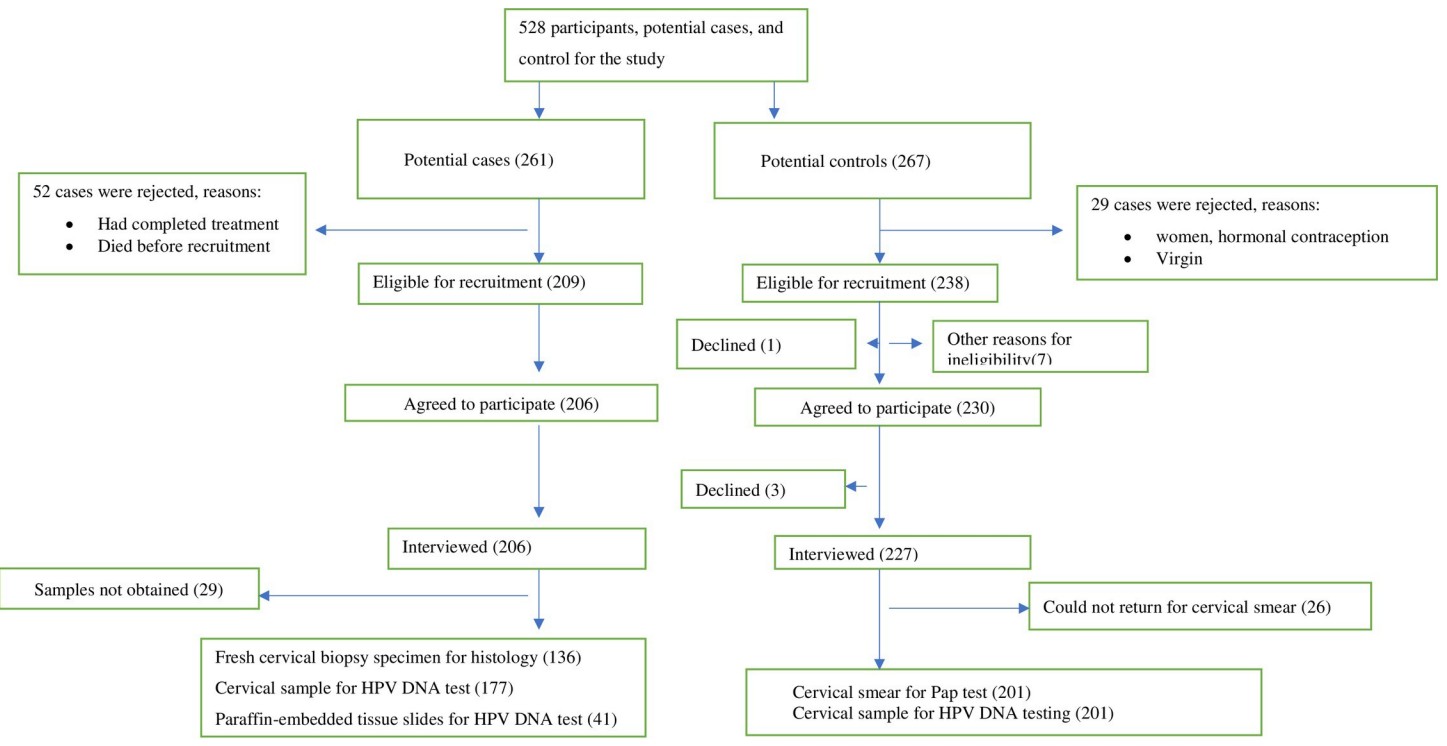

**Fig 1. Summary of the study participants' selection.**

without cervical cancer (all p-value <0.001). The cases and controls did not differ significantly regarding ethnicity and religion.

## HPV DNA positivity

The prevalence of HPV genotypes among those with and those without cervical cancer is shown in Table 2. HPV infections were detected in 84.8% of women with cervical cancer, the majority being the high-risk oncogenic types; HR-HPV infections (89.9%) and LR-HPV infections (10.1%). The most frequently identified HPV genotypes in a descending order of frequency were HPV16 (41.2%), HPV18 (24.9%), HPV45 (16.4%), HPV35 (9.6%) and HPV52 (5.6%). For LR HPV, the non-vaccine HPV genotypes 42, 43 and 44 were more prevalent (7.9%) than the vaccine-type HPV 6/11 (1.1%).

Among women without cervical cancer, the HPV prevalence was 45.8%; HR HPV infection (33.3%) and LR HPV infection (12.4%). The type-specific distribution of single infections, in descending order of frequency was HPV66 (15.4%), HPV52 (14.9%) HPV 35 (6.5%), HPV43 (6.2%), HPV31 (13.5%) and HPV52 (12.8%). With regards to LR HPV, the prevalence of the non-vaccine HPV genotypes 42, 43 and 44 (13%) was six and a half-fold the prevalence of HPV 6/11(2.0%). Non-vaccine types HPV66 and 35 were the most dominant non-vaccine genotypes in women with cervical cancer and those without cervical cancer. HPV 35 was more prevalent among women with cervical cancer, while HPV66 was more prevalent among women without cervical cancer.

Multiple infection with HPV 31 and 33, a vaccine type, were low among women with and those without cervical cancer (Fig 2). The non-vaccine types, HPV 35, 52, and 39, were frequent among women with cervical cancer.

**Table 1. Distribution of baseline characteristics of the subjects.**

| Characteristics | Case | Control |
|---|---|---|
| | n (%) | n (%) |
| **Age group at diagnosis (years)** | | |
| <40 | 12 (6.8) | 138 (68.7) |
| 40–49 | 32 (18.1) | 45 (22.4) |
| 50–59 | 52 (29.4) | 8 (4.0) |
| ≥60 | 81 (45.8) | 10 (5.0) |
| Median (IQR) | 57 (IQR 48.5–70) | 34 (IQR 32–58) |
| **Region of residence** | | |
| Metropolitan | 93 (52.5) | 179 (89.1) |
| Urban | 65 (36.7) | 20 (10.0) |
| Semi-urban | 19 (10.7) | 2 (1.0) |
| **Religion** | | |
| Christian | 152 (85.9) | 181 (90.1) |
| Muslim/other | 25 (14.1) | 19 (9.5) |
| Unknown | 0 (0.0) | 1(0.5) |
| **Ethnicity** | | |
| Ewe | 29 (16.4) | 24 (11.9) |
| Akan | 107 (60.5) | 118 (58.7) |
| Ga-Adangbe | 12 (6.8) | 30 (14.9) |
| Other | 29 (16.4) | 29 (14.4) |
| **Marital status** | | |
| Never Married | 2 (1.1) | 60 (29.9) |
| Married | 83 (46.9) | 116 (57.7) |
| Separated/divorced/widowed | 92 (52.0) | 25(12.4) |
| **Level of education** | | |
| No formal education | 74 (41.8) | 21(10.5) |
| Primary | 35 (19.8) | 24 (11.9) |
| Secondary | 55 (31.1) | 113 (56.2) |
| Tertiary | 12 (6.8) | 41(20.4) |
| Unknown | 1(0.6) | 2(1.0) |
| Other | 35 (19.8) | 24 (11.9) |
| **Occupation** | | |
| Trader | 38 (21.5) | 74 (36.8) |
| Farmer | 31 (17.5) | 5 (2.5) |
| Teacher | 7 (4.0) | 6 (3.0) |
| Hairdresser/seamstress | 2 (1.1) | 34 (16.9) |
| Other | 12 (6.8) | 40 (19.9) |
| None | 82 (46.3) | 38 (18.9) |
| Unknown | 5(2.8) | 4(2.0) |

## Risk of cervical cancer and HPV positivity

Cervical cancer risk was associated with a positive HPV test (AOR 5.78; 95% CI: 2.92–11.42), infection with any of the HR HPV types (AOR: 5.56; 95% CI: 3.27–13.16) and multiple HPV infections (AOR: 9.57 95% CI 4.06–22.56) (Table 3).

**Table 2. The prevalence of vaccine targeted, and non-vaccine targeted human papillomavirus among women with and without cervical cancer in Ghana.**

| HPV type | Case (N = 177) | | | Control (N = 201) | | | Total (N = 378) | | |
|---|---|---|---|---|---|---|---|---|---|
| | Single | Multiple | Total (%) | Single | multiple | Total (%) | Single | Multiple | Total (%) |
| HPV - | - | - | 28 (15.8) | - | - | 109 (54.2) | - | - | 137 (36.2) |
| HPV + | 87 | 62 | 149 (84.2) | 63 | 29 | 92 (45.8) | 150 | 91 | 241 (63.8) |
| **LR HPV+** | 2 | 13 | 15 (10.1) | 8 | 17 | 25 (12.4) | 10 | 30 | 40 (10.6) |
| **HR HPV+** | 85 | 49 | 134 (89.9) | 55 | 12 | 67 (33.3) | 140 | 61 | 201 (53.2) |
| **VT-HPV[1]** | | | | | | | | | |
| HPV18 | 24 | 20 | 44 (24.9) | 0 | 1 | 1 (0.5) | 24 | 21 | 45 (11.9) |
| HPV16 | 41 | 32 | 73 (41.2) | 1 | 1 | 2 (1.0) | 42 | 33 | 75 (20.6) |
| HPV 31 | 0 | 0 | 0 | 0 | 0 | 0 | 0 | 0 | 0 |
| HPV 33 | 0 | 1 | 1 (0.6) | 1 | 0 | 1 (0.5) | 1 | 1 | 2 (0.5) |
| HPV 45 | 9 | 20 | 29 (16.4) | 1 | 4 | 5 (2.5) | 10 | 24 | 34 (9.0) |
| HPV 52 | 0 | 10 | 10 (5.6) | 17 | 13 | 30 (14.9) | 17 | 23 | 40 (10.6) |
| HPV 58 | 0 | 4 | 4 (2.3) | 2 | 0 | 2 (1.0) | 2 | 4 | 6 (1.6) |
| HPV 6/11 | 0 | 2 | 2 (1.1) | 2 | 2 | 4 (2.0) | 2 | 4 | 6 (1.6) |
| Subtotal | 74 | 89 | 163 (92.1) | 24 | 21 | 45 | 98 | 110 | 208 (55.0) |
| **NVT-HPV[2]** | | | | | | | | | |
| HPV 35 | 3 | 14 | 17 (9.6) | 3 | 10 | 13 (6.5) | 6 | 24 | 30 (7.9) |
| HPV 39 | 1 | 7 | 8 (4.5) | 2 | 4 | 6 (3.0) | 3 | 11 | 14 (3.7) |
| HPV 51 | 2 | 7 | 7 (4.0) | 3 | 3 | 6 (3.0) | 3 | 10 | 13 (3.5) |
| HPV 56 | 1 | 4 | 5 (2.8) | 3 | 6 | 9 (4.5) | 4 | 10 | 14 (3.7) |
| HPV 59 | 0 | 3 | 3 (1.7) | 1 | 1 | 2 (1.0) | 1 | 4 | 5 (1.3) |
| HPV 66 | 3 | 4 | 7 (4.0) | 18 | 13 | 31 (15.4) | 21 | 17 | 39 (10.3) |
| HPV 68 | 1 | 5 | 6 (3.4) | 3 | 3 | 6 (3.0) | 4 | 8 | 12 (3.2) |
| HPV 42 | 0 | 2 | 2 (1.1) | 1 | 3 | 4 (2.0) | 1 | 5 | 6 (1.6) |
| HPV 43 | 1 | 7 | 8 (4.5) | 4 | 9 | 13 (6.5) | 5 | 16 | 21 (5.6) |
| HPV 44 | 1 | 3 | 4 (2.3) | 1 | 8 | 9 (4.5) | 2 | 11 | 13 (3.4) |
| Subtotal | 13 | 56 | 69 (37.9) | 39 | 60 | 99 (49.3) | 50 | 116 | 166 (**43.9**) |

HPV: human papillomavirus; LR HPV: low-risk human papillomavirus HR HPV: high-risk human papillomavirus; VT-HPV: vaccine targeted human papillomavirus; NVT-HPV: non-vaccine targeted human papillomavirus virus

### Age-specific HPV infection

Women younger than 40 years of age had more (68.7%) single-type and multiple-type infections (Table 4), with the prevalence, generally declining with age.

## Discussion

Accurate epidemiological information on HPV infections is essential for cervical cancer prevention through HPV DNA testing and vaccination with regional-specific HPV preventive vaccines. The HPV genotypes 16, 18, 45, 35 and 52 were the most common types identified in women with cervical cancer, and in decreasing order, genotypes 66, 52, 35, 56 and 45 were prevalent in women without cancer. High-risk HPV infections are highly prevalent among Ghanaian women, with an estimated cervical cancer risk of about six- to ten-fold in a positive HPV test.

The case differed significantly from the controls regarding age, residence, marital status, level of education and occupation. The risk of cervical cancer is higher in rural disease [28].

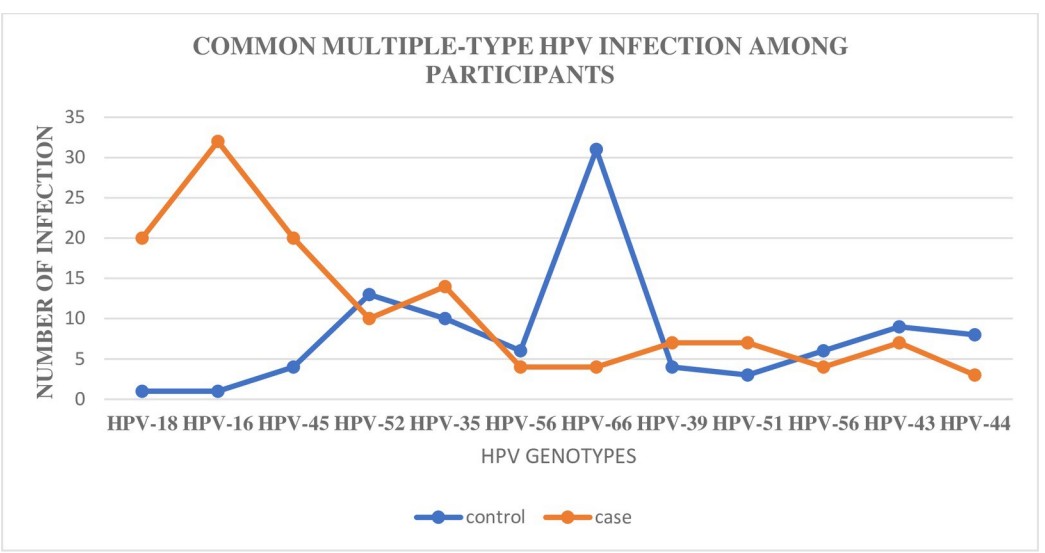

**Fig 2. Pattern of common HPV multiple infections in women with and without cervical cancer.** *Note*: HPV: Human papillomavirus.

Most women with cervical cancer were likely to have been referred from primary health care centres in their towns and communities. However, controls were women seeking healthcare for reasons other than cervical cancer. Therefore, controls were likely to live in areas where the tertiary institutions are situated, which are metropolitan.

The HPV genotypes distribution, as observed, is concordant with sub-regional data [13]. The differences in HPV infection profile between women with and without the cancer is dependent on the ability of the immune system to eliminate the infection. The HPV infection profile at any point represents a cocktail of persistent and new infections. Krings et al. reported 6.7% and 21.2% rates of new and old HPV infections respectively, among in a cohort of healthy women after a 4-years of residence in Ghana [29]. HPV 16 and 18, despite their low infection rate among the healthy individuals, were well represented among the cases, confirming their ability to resist clearance. HPV45 showed a similar trend to HPVs 16 and 18.

The bivalent vaccines (containing type 16 and 18) have exhibited significant cross-protection efficacy against CIN from infections of type HPV31 and HPV33 [30]. It is interesting to

**Table 3. Risk of cervical cancer by HPV infection.**

| Characteristic | Cases (N = 177) | Control (N = 201) | Total (N = 378) | COR (95% CI) | AOR (95% CI) |
|---|---|---|---|---|---|
| | n (%) | n (%) | n (%) | | |
| **HPV DNA positivity** | | | | | |
| HPV- | 27 (15.3) | 109 (54.2) | 136 (36.0) | 1 | 1 |
| HPV+ | 150 (84.8) | 92 (45.8) | 242 (64.0) | 6.58 (4.01–10.80) | 5.78 (2.92–11.42) |
| **Multiple HPV types** | | | | | |
| HPV- | 27 (15.3) | 109 (54.2) | 136 (36.0) | 1 | 1 |
| HPV+ (1 type) | 88 (49.7) | 63 (31.3) | 151 (40.0) | 5.64 (3.32–9.59) | 4.48 (2.16–9.27) |
| HPV+ (>1 types) | 62 (35.0) | 29 (14.4) | 91 (24.1) | 8.63 (4.69–15.88) | 9.57 (4.06–22.56) |

AOR: Adjusted odds ratio; CI confidence interval, COR: Crude odds ratio; HPV: Human papillomavirus; DNA: deoxyribonucleic acid.

The model included positive HPV DNA test and multiple infection. Cervical cancer risk was associated with a positive HPV test, infection with any of the high-risk HPV types and multiple HPV infections.

**Table 4. Age-specific prevalence of HPV DNA in control women.**

| | HPV+ (1 type) | HPV+ (2 type) | HPV+ (3 types) | HPV+ (>3 types) | Total |
|---|---|---|---|---|---|
| | (N = 63) | (N = 15) | (N = 15) | (N = 15) | (N = 201) |
| | n (%) | n (%) | n (%) | n (%) | n (%) |
| Age group (years) | | | | | |
| <40 | 37 (58.7) | 13 (86.7) | 4 (66.7) | 6 (75.0) | 138 (68.7) |
| 40–49 | 19 (30.2) | 1 (6.7) | 2 (33.3) | 1 (12.5) | 45 (22.4) |
| 50–59 | 3 (4.8) | 0 (0.0) | 0(0.0) | 0(0.0) | 8 (4.0) |
| ≥60 | 4 (6.4) | 1 (6.7) | 0(0.0) | 1(12.5) | 10 (5.0) |

HPV: Human papillomavirus

note that none of the surveyed women tested positive for the HPV 31, a vaccine-targeted HPV. Low infection rates from this genotype have also been reported in earlier studies [29, 31]. This supports the suggestion that HPV31 infections rapidly clear after primary contact, reducing its carcinogenic potential [11]. A similar clearance pattern is seen in HPV 66 infections, particularly in young women [32], which might account for a relatively lower infection rate of the genotype among the cases despite being the highest HPV infection detected in the women without cancer. Infections with HPV35, a nonvaccine genotype, were common irrespective of group, suggesting their possible role in cervical carcinogenesis in Ghana is unclear. Several studies in the sub-region have documented significant levels of HPV 35 infections in women from a wide spectrum of the female population [25, 33–35]. Scanty data exist on low-risk HPV infection in Sub-Saharan Africa. The prevalence of condyloma associated low-risk HPV was higher (HPV 6/11 versus HPV 43/44, 2.0% and 11%, respectively) than the average rate in West Africa (2%) and high compared to rates reported in the general population in Southern Africa (4%) and in Eastern Africa (8%). In this study, non-vaccine low-risk types, 43 and 44 featured more prominently than HPV 6/11. These dynamics in low-risk HPV prevalence and distribution may be influenced by lack of systematic HPV vaccination programmes.

Cervical cancer is essentially infective in origin, and persistent high-risk human papillomavirus (HPV) infection is central in tumorigenesis [36, 37]. Co-infection with multiple HPV types is associated with longer infection duration and a higher risk for cervical cancer [38, 39]. Multiple-type infections involving HPV 16, 18, 35, 52 and 45 are likely to persist and elude clearance by the immune system, a prerequisite for the development of the cancer [29, 40–42]. In the current study, compared to participants with a negative test, those who tested positive to any high-risk genotype had a 5-10-fold increase in the risk of cervical cancer. HPV 16 and 18, like in other regions, were found to be the main contributors to cervical carcinogenesis in Ghana.

Knowledge of genotypic distribution in a specific population has implications for vaccine choice and predicted impact. Largely, the protection afforded by the HPV vaccine is type-specific [18]. The results of this study indicate that combating persistent infection of HPV 16, 18, 45, 35 and 52 will be essential in consideration for regional specific HPV vaccines and DNA screening tests. The results suggest that 66.1% and 67.2% of cervical cancer diagnosed in Ghana could be prevented with widespread licensed bivalent and quadrivalent vaccination, respectively. Nonavalent vaccine, provides additional cover for HPV 31, 33, 45,52 and 58, with an estimated broader protection against cervical cancer (92.1%).

Age is one of the most important risk factors of HPV infection [43]. The prevalence at different ages, is critical for designing age-specific prophylactic HPV vaccination program. Our study observed two peaks in HPV prevalence. This observation is consistent with available data in Ghana [8, 44]. The highest prevalence was observed in women younger than 40 years.

Non-existence of a national or systematic vaccination programme explains the high overall prevalence of the infection.

HPV vaccination of young adolescents in Ghana is possible via the established infrastructure and logistics system provided by the Expanded Program on Immunization (EPI). Initiation of HPV vaccination for young adolescents aged 10–14 years may be appropriate to include most HPV naive young girls [16]. A cervical cancer educational and vaccination program (targeted at upper primary and early junior high schools) could be implemented as part of a national cervical cancer control program. Multiple factors, including host susceptibility to HPV [45] and senescence probably explain the second peak, seen in women older than 60 years. Prioritizing screening for women 50 years and above when resources are limited may be worthwhile because the median age of women with cervical cancer in this study was 57 years, and a third of all the cancers were diagnosed in those aged 50 and above. However, HPV DNA based testing in women aged 30 years and above is the most preferred and widely used cervical cancer screening test globally [46].

The major strength of this study was that it provided the comparative distribution and prevalence of HPV genotype in women with and without cervical cancer at the two largest cervical cancer treatment centres in the country. Information from these centres may have a significant role in decisions regarding the national cervical cancer prevention programmes, i.e., HPV vaccination and HPV DNA testing for the control of cervical cancer in Ghana.

Our study had a few limitations. HPV genotype prevalence and distribution may differ among regions within the same country. Although most cervical cancer diagnosed and treated in Ghana are managed at Korle Bu and Komfo Anokye Teaching Hospitals, this information may not be generalizable to the entire cervical cancer population or the HPV infection burden of the general female population. Nevertheless, our HPV infection profile of women studied is concordant with many studies in Ghana and other countries in the sub-region. Three specimens were collected: exfoliated cervical cells, fresh biopsy, and paraffin-embedded samples. HPV genotyping from a cervical sample is influenced by specimen type [47, 48]. However, a validation study comparing HPV DNA detection rates from exfoliated cervical cells and biopsy specimens from the same subject did not detect a significant difference [49]. Other studies have also reported high levels of agreement between paraffin-embedded tissue with other types of samples. A disadvantage of paraffin-embedded samples is the risk of degradation of DNA and the possibility of a negative test as the sample ages. This phenomenon was unlikely as the specimens were from women newly diagnosed with cervical cancer. HPV DNA detection assays vary in their sensitivity and ability to detect multiple HPV types [50]. The sensitivity of the nested multiplex assay is comparable to that of the PGMY09/11 assay, which is considered one of the most reliable assays for HPV DNA detection and typing. The nested multiplex PCR assay may not detect every HPV genotype associated with cervical cancer, but this is considered unlikely.

While HPV vaccination can be expected to reduce the incidence of cervical cancer in the long-term, it is necessary to reduce the risk of cervical cancer in those already infected. Therefore, on-going cervical screening will be essential to reducing the burden of cervical cancer in Ghana for several decades to come. Currently, Ghana does not have a national HPV screening program for cervical cancer. However, there are some public and private health facilities that offer such services. The current study provides useful information on the type of HPV likely to lead to cervical cancer and may help in the triage of women for follow-ups.

## Conclusion

Women with cervical cancer in Ghana have HPV infection with multiple genotypes, including some non-vaccine genotypes, with an estimated cervical cancer risk of about six- to ten-fold in

the presence of a positive HPV test. HPV DNA testing for cervical screening and multivalent vaccine targeted at the most prevalent HPV types will be a valuable contribution to improve Ghana's cervical cancer control programme than the currently available licensed vaccine. Large population-based studies are required in countries where cervical cancer is most prevalent to determine non-vaccine HPV genotypes which should be considered for the next-generation.

## Supporting information

**S1 File.**
(DOCX)

## Acknowledgments

We would like to thank the participants of this study.

## Author Contributions

**Conceptualization:** Yvonne Nartey, Philip C. Hill, Brian Cox.

**Data curation:** Yvonne Nartey, Kwabena Amo-Antwi, Philip C. Hill, Richard H. Asmah, Joel Yarney, Brian Cox.

**Formal analysis:** Yvonne Nartey, Kwabena Amo-Antwi, Brian Cox.

**Funding acquisition:** Brian Cox.

**Investigation:** Yvonne Nartey, Philip C. Hill, Nelson Damale, Brian Cox.

**Methodology:** Yvonne Nartey, Kwabena Amo-Antwi, Philip C. Hill, Edward T. Dassah, Richard H. Asmah, Kofi M. Nyarko, Ramatu Agambire, Thomas O. Konney, Joel Yarney, Nelson Damale, Brian Cox.

**Project administration:** Yvonne Nartey, Kwabena Amo-Antwi, Philip C. Hill, Richard H. Asmah, Joel Yarney, Nelson Damale, Brian Cox.

**Resources:** Philip C. Hill, Richard H. Asmah, Joel Yarney, Brian Cox.

**Software:** Yvonne Nartey, Brian Cox.

**Supervision:** Yvonne Nartey, Kwabena Amo-Antwi, Philip C. Hill, Richard H. Asmah, Joel Yarney, Nelson Damale, Brian Cox.

**Validation:** Yvonne Nartey, Kwabena Amo-Antwi, Philip C. Hill, Edward T. Dassah, Richard H. Asmah, Kofi M. Nyarko, Ramatu Agambire, Thomas O. Konney, Joel Yarney, Brian Cox.

**Visualization:** Yvonne Nartey, Kwabena Amo-Antwi, Philip C. Hill, Edward T. Dassah, Richard H. Asmah, Kofi M. Nyarko, Ramatu Agambire, Thomas O. Konney, Joel Yarney, Nelson Damale, Brian Cox.

**Writing – original draft:** Yvonne Nartey, Kofi M. Nyarko.

**Writing – review & editing:** Yvonne Nartey, Kwabena Amo-Antwi, Philip C. Hill, Edward T. Dassah, Richard H. Asmah, Kofi M. Nyarko, Ramatu Agambire, Thomas O. Konney, Joel Yarney, Nelson Damale, Brian Cox.

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
