## [Decision Letter · Decision Letter 0]

14 Nov 2022

PONE-D-22-28450Human papillomavirus genotype distribution among women with and without cervical cancer: Implication for vaccination and screening in Ghana.PLOS ONE

Dear Dr. Yvonne Nartey,

Thank you for submitting your manuscript to PLOS ONE. After careful consideration, we feel that it has merit but does not fully meet PLOS ONE’s publication criteria as it currently stands. Therefore, we invite you to submit a revised version of the manuscript that addresses the points raised during the review process.

We look forward to receiving your revised manuscript.

Kind regards,

Gulzhanat Aimagambetova

Academic Editor

PLOS ONE

3. Our staff editors have determined that your manuscript is likely within the scope of our Early Detection, Screening and Diagnosis of Cancer Call for Papers. This editorial initiative is headed by in-house PLOS editors. This Call for Papers aims to explore recent advances in the early detection of cancer and implications of these advances for patient survival. Additional information can be found on our announcement page: https://collections.plos.org/call-for-papers/early-detection-screening-and-diagnosis-of-cancer/

If you would like your manuscript to be considered for this collection, please let us know in your cover letter and we will ensure that your paper is treated as if you were responding to this call. Please note that being considered for the Call for Papers does not require additional peer review beyond the journal’s standard process and will not delay the publication of your manuscript if it is accepted by PLOS ONE. If you would prefer to remove your manuscript from collection consideration, please specify this in the cover letter.

4. Please include a complete copy of PLOS’ questionnaire on inclusivity in global research in your revised manuscript. Our policy for research in this area aims to improve transparency in the reporting of research performed outside of researchers’ own country or community. The policy applies to researchers who have travelled to a different country to conduct research, research with Indigenous populations or their lands, and research on cultural artefacts. The questionnaire can also be requested at the journal’s discretion for any other submissions, even if these conditions are not met.  Please find more information on the policy and a link to download a blank copy of the questionnaire here: https://journals.plos.org/plosone/s/best-practices-in-research-reporting. Please upload a completed version of your questionnaire as Supporting Information when you resubmit your manuscript.

Reviewers' comments:

Reviewer's Responses to Questions

**Comments to the Author**

1. Is the manuscript technically sound, and do the data support the conclusions?

Reviewer #1: Yes

Reviewer #2: Yes

2. Has the statistical analysis been performed appropriately and rigorously? 

Reviewer #1: Yes

Reviewer #2: Yes

3. Have the authors made all data underlying the findings in their manuscript fully available?

Reviewer #1: Yes

Reviewer #2: Yes

4. Is the manuscript presented in an intelligible fashion and written in standard English?

Reviewer #1: Yes

Reviewer #2: Yes

5. Review Comments to the Author

Reviewer #1: This is an well written article on the topic of HPV in Ghana. The analysis presented in the paper are really interesting. I would leave a minor comment on enriching the discussion. In the last paragraph, you might include two or three sentences on Ghana's existing HPV screening program and how your study findings can improve the existing approach.

Reviewer #2: Nartey and colleagues present an interesting study on the prevalence of different HPV genotypes among women in Ghana. Studies like this are particularly important in understanding if newer vaccines are needed to treat different strains and the effectiveness of HPV vaccination in other areas. Additional comments are noted below.

*In line 109, authors state that HPV vaccine effectiveness studies have mainly been conducted among Caucasians. While there is a lack of diversity in research participants in biomedical research, this statement ignores the positive gains made in preventing cervical cancer in multiple countries that has had benefits for people across multiple races. Additionally, this statement seems out of place considering the authors do not mention race and ethnicity throughout the article. The authors should consider either removing this statement are include additional information about how research is needed among different races/ ethnicities and results from this study differ from previous studies conducted among majority Caucasian populations.

*It's unclear if the authors are trying to promote the usage of currently available HPV vaccines in Ghana or if they are requesting a new one. It would be helpful if the authors were clearer about this. Especially in lines 470-471 in the conclusion.

6. PLOS authors have the option to publish the peer review history of their article (what does this mean?). If published, this will include your full peer review and any attached files.

Reviewer #1: **Yes: **Abdullah Nurus Salam Khan

Reviewer #2: No

---

## [Author Response · Author response to Decision Letter 0]

15 Dec 2022

22nd November, 2022

Dear Dr Aimagambetova,

We thank you for taking the time to carefully read our manuscript and for the valuable comments you have provided, which helped us in improving the revised paper that we are re-submitting. We have made all requested changes, ensured it complies with all the journal requirements.

Yours truly,

Dr Yvonne Nartey

Here is our point-by-point response to the reviewer’s comments: 

(Reviewer comments in bold)

This is an well written article on the topic of HPV in Ghana. The analysis presented in the paper are really interesting. I would leave a minor comment on enriching the discussion. In the last paragraph, you might include two or three sentences on Ghana's existing HPV screening program and how your study findings can improve the existing approach.

Response: We have, accordingly, added the following sentence to the last paragraph of the discussion session.

“Currently, Ghana do not have a national HPV screening program for cervical cancer. However, there are some public and private health facilities that offer such services. The current study provides useful information on the type of HPV likely to lead to cervical cancer and may help in the triage of women for follow-ups. “ - Page 20, paragraph 1, line 2.

Nartey and colleagues present an interesting study on the prevalence of different HPV genotypes among women in Ghana. Studies like this are particularly important in understanding if newer vaccines are needed to treat different strains and the effectiveness of HPV vaccination in other areas. Additional comments are noted below.

In line 109, authors state that HPV vaccine effectiveness studies have mainly been conducted among Caucasians. While there is a lack of diversity in research participants in biomedical research, this statement ignores the positive gains made in preventing cervical cancer in multiple countries that has had benefits for people across multiple races. Additionally, this statement seems out of place considering the authors do not mention race and ethnicity throughout the article. The authors should consider either removing this statement are include additional information about how research is needed among different races/ ethnicities and results from this study differ from previous studies conducted among majority Caucasian populations.

Response: We have, accordingly, revised the sentence to reflect the reviewer comment. The sentence has been altered to the following: 

“The available HPV vaccines have activity against up to nine HPV types (6, 11, 16,18, 31 33 45 52 58)”. – Page5, paragraph 2, line 7.

It's unclear if the authors are trying to promote the usage of currently available HPV vaccines in Ghana or if they are requesting a new one. It would be helpful if the authors were clearer about this. Especially in lines 470-471 in the conclusion.

Response: We have, accordingly, revised the sentence to reflect the reviewer comment. The sentence has been altered to the following: 

“HPV DNA testing for cervical screening and multivalent vaccine targeted at the most prevalent HPV types will be a valuable contribution to Ghana’s cervical cancer control programme than the currently available licensed vaccine.” – page 20, paragraph 2, line 8.

---

## [Decision Letter · Decision Letter 1]

2 Jan 2023

Human papillomavirus genotype distribution among women with and without cervical cancer: Implication for vaccination and screening in Ghana.

PONE-D-22-28450R1

Dear Dr. Yvonne Nartey,

We’re pleased to inform you that your manuscript has been judged scientifically suitable for publication and will be formally accepted for publication once it meets all outstanding technical requirements.

Kind regards,

Gulzhanat Aimagambetova

Academic Editor

PLOS ONE

Reviewers' comments:

Reviewer's Responses to Questions

**Comments to the Author**

1. If the authors have adequately addressed your comments raised in a previous round of review and you feel that this manuscript is now acceptable for publication, you may indicate that here to bypass the “Comments to the Author” section, enter your conflict of interest statement in the “Confidential to Editor” section, and submit your "Accept" recommendation.

Reviewer #1: All comments have been addressed

Reviewer #2: All comments have been addressed

2. Is the manuscript technically sound, and do the data support the conclusions?

Reviewer #1: Yes

Reviewer #2: Yes

3. Has the statistical analysis been performed appropriately and rigorously? 

Reviewer #1: Yes

Reviewer #2: Yes

4. Have the authors made all data underlying the findings in their manuscript fully available?

Reviewer #1: Yes

Reviewer #2: Yes

5. Is the manuscript presented in an intelligible fashion and written in standard English?

Reviewer #1: Yes

Reviewer #2: Yes

6. Review Comments to the Author

Reviewer #1: This is a very important paper on the topic. Authors have addressed the comments and feedback adequately.

Reviewer #2: The authors did a great job in addressing reviewer comments! I have no additional comments or concerns about this publication.

7. PLOS authors have the option to publish the peer review history of their article (what does this mean?). If published, this will include your full peer review and any attached files.

Reviewer #1: **Yes: **Abdullah Nurus Salam Khan

Reviewer #2: No

---

## [Editor Report · Acceptance letter]

6 Jan 2023

PONE-D-22-28450R1 

Human papillomavirus genotype distribution among women with and without cervical cancer: Implication for vaccination and screening in Ghana. 

Dear Dr. Nartey:

I'm pleased to inform you that your manuscript has been deemed suitable for publication in PLOS ONE. Congratulations! Your manuscript is now with our production department. 

Kind regards, 

on behalf of

Dr. Gulzhanat Aimagambetova 

Academic Editor

PLOS ONE